# Lung and Gut Microbiome in COPD

**DOI:** 10.3390/jpm13050804

**Published:** 2023-05-08

**Authors:** Efstathios Karakasidis, Ourania S. Kotsiou, Konstantinos I. Gourgoulianis

**Affiliations:** 1Department of Respiratory Medicine, School of Health Science, University of Thessaly, Biopolis, 41110 Larissa, Greece; 2Department of Human Pathophysiology, Faculty of Nursing, School of Health Science, University of Thessaly, Gaiopolis, 41110 Larissa, Greece

**Keywords:** COPD, exacerbations, gut, lung, microbiome

## Abstract

Chronic obstructive pulmonary disease (COPD) is one of the leading causes of death worldwide. The association between lung and gut microbiomes in the pathogenesis of COPD has been recently uncovered. The goal of this study was to discuss the role of the lung and gut microbiomes in COPD pathophysiology. A systematic search of the PubMed database for relevant articles submitted up to June 2022 was performed. We examined the association between the lung and gut microbiome dysbiosis, reflected in bronchoalveolar lavage (BAL), lung tissue, sputum, and feces samples, and the pathogenesis and progression of COPD. It is evident that the lung and gut microbiomes affect each other and both play a vital role in the pathogenesis of COPD. However, more research needs to be carried out to find the exact associations between microbiome diversity and COPD pathophysiology and exacerbation genesis. Another field that research should focus on is the impact of treatment interventions targeting the human microbiome in preventing COPD genesis and progression.

## 1. Introduction

According to the 2022 GOLD report, COPD is a “common, preventable and treatable disease characterized by persistent respiratory symptoms and airflow limitation due to airway and/or alveolar abnormalities” [1]. COPD accounted for 6% (3.23 million) of deaths in 2019, making it the third leading cause of death worldwide [2]. Moreover, it is predicted that in the following years, the prevalence and mortality of COPD will continue to increase so that by 2060, COPD will be the cause of over 5.4 million deaths per year and will be the leading cause of death among other non-communicable diseases [1]. A difficult aspect in the follow-up and treatment of patients with COPD is that COPD exacerbations are associated with a decrease in the patients’ quality of life, health status, and the progression of the disease by increasing the rate of decline of lung function. Exacerbations are responsible for frequent medical visits, hospital admissions, and changes in medication, and are the leading cause of death in patients with COPD [3,4,5,6]. 

The human microbiome is the sum of all forms of microorganisms and their genome inhabiting an individual at a given time [7]. These microorganisms can be found predominantly in the gut and other mucosal surfaces of the human body, such as the skin, oral cavity, airways, and vagina [8,9]. Dysbiosis is any change in the composition of resident commensal communities relative to the community found in healthy individuals [10]. The low diversity of the microbiome is considered a marker of dysbiosis [11], while high diversity has been generally associated with health and temporal stability [10,12,13].

A better understanding of the etiology of COPD could provide a stimulating source of information concerning the prevention and treatment of the disease and its exacerbations. The primary risk factor for COPD is active and passive cigarette smoking [4,14]. Smoking alters both gut and lung microbiomes by changing the microecological structure of the mouth via several ways [5,7,13]. Smoke affects the microenvironment of bacteria (pH, oxygen tension, acid production) that results in changes in the microbiome [15]. Furthermore, tobacco contains microbes that inhabit humans via smoking. Cigarettes also induce biofilm formation and the impairment of the immune system. Both of these enable the proliferation and colonization of specific bacterial taxa [16,17]. Morris et al. found out that *Porphyromonas*, *Neisseria*, and *Gemella* were diminished in the mouth microbiome of smokers [18]. Wang et al. found that *Haemophilus* was more prevalent in smokers versus nonsmokers [19]. Apart from smoking, environmental factors, such as air pollution and occupational exposure to dust, chemicals, fumes, and gases are risk factors for COPD development [4,5].The microbiome can be altered also by environmental factors (air pollution and allergens) and becomes less diverse with age [7,20].

Interestingly, a Western diet has been associated with the impairment of lung function and increased mortality in COPD patients. In contrast, a diet rich in antioxidants improves lung function, thereby reducing mortality due to COPD [13,21]. Diet plays a vital role in preserving diversity in the gut microbiome [10,11,20,22], and the consumption of a low-fiber diet causes a loss of gut bacterial diversity [7,8]. Another factor that contributes to dysbiosis is the intake of drugs, and the most important are antibiotics [7,10,11,20,23] that act by directly inhibiting bacterial growth [24]. Other common drugs, such as proton-pump inhibitors, laxatives, and metformin, are associated with changes in the microbiomes, especially the gut microbiome. Inhaled corticosteroids or inhaled bronchodilators also affect the lung microbiome [9,24,25]. 

Changes in the microbiomes have been studied mainly in gastrointestinal diseases, such as IBD, but in the last few years, interest in the relationship between microbiomes and other common airway diseases, such as COPD and asthma has increased [26]. Although a relationship between COPD and microbiomes is established, it is not known whether dysbiosis causes the disease or is a result of the disease [7]. In that context, this review aims to present all of the current knowledge regarding the linkage of lung and gut microbiomes in the context of COPD pathophysiology and how this association has expanded to the prevention and treatment of COPD and its exacerbations.

## 2. An Overview of the Human Gut Microbiome

The gut microbiome is the most abundant in the microorganism population and consists of approximately 1 × 10^14^ microorganisms belonging to more than a thousand bacterial species [27]. The composition of the gut microbiome is different through the gastrointestinal (GI) tract due to variations in oxygen gradients and pH [7,28].The six most prevalent phyla in a healthy gut microbiome are Firmicutes, Bacteroidetes, Fusobacteria, Actinobacteria, Proteobacteria, and Verrucomicrobia. Firmicutes and Bacteroidetes represent 90% of phyla in the gut microbiome [7,13,20,22,27]. The most predominant classes and genera from these six phyla are *Bifidobacterium*, *Lachnospiraceae*, *Streptococcus*, *Enterococcus*, and *Lactobacillus* [13].

The gut microbiome has various roles in the human body, as it is involved in innate and adaptive immunity, intestinal immunity, and metabolic processes [7,9,11,27,28]. The gut microbiome achieves these effects via the production of metabolites, the most well-known of which are short-chain fatty acids (SCFAs), such as acetate, propionate, and butyrate. SCFAs are produced through the fermentation of indigestible oligosaccharides [9], they have anti-inflammatory properties [13], help preserve colonic epithelial integrity, and regulate the host energy balance [29]. The role of the two most prevalent phyla of the gut microbiome is Firmicutes bacteria, via the synthesis of SCFAs, which are involved in the nutrition and metabolism of the host, while Bacteroidetes bacteria are associated with immunomodulation [20].

## 3. An Overview of the Human Lung Microbiome

The lung microbiome represents 6% of the total microbiomes in the body and its biomass is significantly lower than that of the gut [30]. The main source of the lung microbiome is the micro aspiration of oral flora [25,31,32]. Coughing and mucociliary clearance, as well as local conditions for microbial growth, such as pH and oxygen concentration, can influence the composition of the lung microbiome [31]. Similar to the gut microbiome, the lung microbiome predominantly consists of the phyla Bacteroidetes, Firmicutes, and Proteobacteria, followed by Actinobacteria [7,31,32,33,34]. At the genus level, *Streptococcus* is the most common [9,32,33,34,35] followed by *Prevotella* [9,32,34,35], *Fusobacterium* [9,35], and *Veillonella* [32,33,34]. 

## 4. The Gut–Lung Theory

In the last few years, many studies have shown that there is an interaction between the lungs and the gastrointestinal tract, and this is known as the gut–lung theory. This theory supports that because of their homologous embryonic development from the endoderm, the intestines and lungs are linked with each other, and there is a “common mucosal response”, where the gastrointestinal mucosa may regulate immune responses in lungs and vice versa [7,36]; therefore, the mucosa influences the development of lung or gut diseases [9,31].

The interaction between these two organs includes the transportation of microbes from one organ to the other and the transportation of metabolites mainly from the gut microbiome to the lungs. The most well-known metabolites transported through the gut–lung axis are SCFAs that play an important role in local and systematic immune responses as well as in tissue homeostasis [30,31,37]. The exact mechanism by which SCFAs act in the lungs is not yet known. Dang et al. proposed that SCFAs do not act directly in the lungs because they cannot accumulate in the lungs, and lung bacteria cannot produce in substantial amounts. Alternatively, SCFAs are produced by the gut microbiome and after reaching the systematic circulation, they activate immune cells in the periphery that are then are recruited to the lungs. Furthermore, SCFAs prime myeloid cells in the bone marrow, which then migrate to the lungs and modulate the immune response [37]. 

## 5. COPD and Microbiomes

Smoking is the predominant risk factor for COPD, but only some smokers develop COPD. Why this happens is unknown, but new techniques in the culture of microorganisms have shed some light on the possible link between the gut and lung microbiomes and COPD pathogenesis [9].

Because of the gut–lung axis hypothesis, interest into whether lung and gut microbiomes relate to the pathogenesis of COPD has increased. The relationship between the lung microbiome reflected by either sputum, bronchoalveolar lavage fluid (BALF), or lung tissue and the gut microbiome reflected by fecal samples and the pathogenesis of COPD has been recently presented [38].

There is a theory known as the vicious circle hypothesis for the pathogenesis of COPD, and it states that lung dysbiosis provides the constant inflammatory stimulus in COPD [35]. In particular, cigarette smoke can influence the immune defense and cause lung dysbiosis that causes maladaptive inflammatory responses, impairment of the lung defense and, therefore, lung dysbiosis, resulting in the development of a vicious circle [38]. 

The studies that examined the role of microbiomes in COPD pathophysiology by analyzing lung tissue, BAL, sputum, and feces samples in COPD patients in comparison with healthy controls are summarized in Table 1.

## 6. Lung Microbiome in COPD Patients According to Lung Tissue Samples

Sze et al. examined the microbiome in lung tissues obtained from patients with very severe COPD, compared them with those of smokers and non-smokers without COPD and patients with cystic fibrosis, and their results supported that severe COPD was associated with an increase in the phylum Firmicutes, which is attributable to an increase in abundance of *Lactobacillus* as well as an increase in abundance of *Burkholderia* genus [39]. Erb-Downward et al. found that the lung microbiome differs in different regions of the same lung showing that local factors may play a substantial role in the dominance of some microorganisms against others [35]. In another study, Sze et al. examined the lung tissues from five patients with COPD GOLD 4 and compared them with the lung tissues of four donors. They observed that the abundance of both Proteobacteria and Actinobacteria was increased in samples obtained from COPD patients. At the same time, Firmicutes and Bacteroidetes phyla were decreased in COPD samples compared to the lung tissues of donors. They pointed out that these differences were associated with COPD pathogenesis and progression. They also supported the idea that ten species can be used to discriminate microbiomes between controls and patients with COPD GOLD stage 4. These species were *Prevotella oralis*, *Streptococcus*, *Prevotella oris*, *Porphyromonas*, *Flavobacterium succinicans*, *Haemophilus influenza*, *Bacteroidales*, *Elizabethkingia meningoseptica*, *Dialister*, and *Flavobacterium gelidilacus* [41]. However, it should be noted that only patients with GOLD stage 4 had been recruited in the studies mentioned above; thus, the microorganisms found in high abundance in COPD lungs might be associated with the disease’s development and/or progression [39]. Pragman et al. examined the lung tissue and nasal, bronchial, and oral samples of nine patients who underwent lobectomy for suspected or confirmed lung cancer. Patients that used antibiotics or oral corticosteroids in the last two months were excluded. Three of them used inhaled corticosteroids. Their results supported that the upper respiratory bronchial tree is richer but less diverse in microorganisms than a lower bronchial tree. They supported that the most common genera found in COPD patients was *Streptococcus*. Other microorganisms that were commonly found included *Corynebacterium*, *Alloiococcus*, *Prevotella*, *Veillonella*, *Rothia*, *Neisseria*, and *Staphylococcus* [40].

## 7. Lung Microbiome in COPD Patients According to BALF Samples

The most prevalent species in BALF samples of COPD patients are Actinobacteria, followed by Firmicutes, Proteobacteria, Nitrospira, Fusobacteria, and Bacteroidetes, [25,30,32,45], while the most common genera are *Prevotella*, *Pseudomonas*, *Fusobacterium*, *Veillonella*, *Streptococcus*, *Haemophilus*, *Lactobacillus*, *Veillonella*, and *Bacillus* [30,32,35,45]. Zakharkina et al. examined the BALF of nine healthy patients and compared them with nine patients with COPD, and found that *Afipia*, *Brevundimonas*, *Curvibacter*, *Moraxella*, *Neisseria*, *Undibacterium*, *Corynebacterium*, *Capnocytophaga*, and *Leptolyngbia* were the most common genera that were increased in COPD patients [30]. Pragman et al. examined the BAL of 22 patients with moderate or severe COPD compared with ten healthy controls. They found that the samples from severe COPD patients contained more Firmicutes and fewer Actinobacteria and Proteobacteria than those from patients with moderate COPD, but this difference was not statistically significant. Furthermore, they pointed out that the development of COPD was associated with a significant increase in microbial diversity [25], which is in contrast with the conclusion of Erb et al., who found that the impairment of lung function and the progression of COPD was related to the reduction in microbial diversity and domination by *Pseudomonas* spp. in BALF samples [35]. These contradictory results may be attributed to the fact that both studies lack a substantial sample size, as well as that patients with advanced COPD need more medications and frequent hospitalizations that might impact their microbiome composition and, therefore, microbial diversity. Pragman et al. supported that the increased microbial diversity is associated with the increased age of patients suffering from advanced COPD rather than the progression of the disease [25].

## 8. Lung Microbiome in COPD Patients According to Sputum Samples

The most commonly observed species in sputum samples from patients with COPD are Firmicutes, Proteobacteria, Bacteroidetes, Actinobacteria, and Fusobacteria [19,33,42,43,44,45]. At the genus level, the most abundant genera in sputum samples of COPD patients are *Streptococcus*, *Pseudomonas*, *Haemophilus*, *Neisseria*, *Moraxella*, *Prevotella*, *Veillonella*, and *Rothia*. Wu et al. examined the sputum samples from ten patients with COPD and ten healthy volunteers. They documented a relative abundance of *Streptococcus pneumoniae*, *Klebsiella pneumoniae*, and *Pseudomonas aeruginosa* in patients with COPD. They also emphasized that patients with COPD had an increased diversity in their sputum compared to healthy individuals [6].

In contrast, Wang et al. examined the sputum samples from 16 healthy people compared with 43 patients with COPD, and suggested a reduced diversity in sputum, which was related to COPD pathogenesis [19]. Wang et al. also found an increase in the relative abundance of *Moraxella*, *Streptococcus*, and *Actinobacteria* [19]. In another study, Dicker et al. examined the sputum samples of 253 clinically stable COPD patients, and found that low microbiome diversity and a high abundance of Proteobacteria and Haemophilus were associated with the low eosinophil count in peripheral blood. In contrast, the high abundance of Firmicutes and *Streptococcus* was associated with an increased eosinophil count in peripheral blood [42]. This finding is important given that blood eosinophil counts are used as a biomarker to help guide inhaled corticosteroids used in clinical practice. Apart from playing a role in the pathogenesis of COPD, it seems that the lung microbiome was associated with COPD severity. Decreased microbiome diversity, increased abundance of *Proteobacteria*, *Firmicutes*, *Pseudomonas* spp., and *Haemophilus*, and decreased numbers of *Actinobacteria*, *Prevotella*, and *Veillonella* were related to increased disease severity [7,25,35,39,50]. Sputum neutrophil counts were positively correlated with *Moraxella*, *Haemophilus*, and *Neisseria*, and were negatively correlated with *Streptococcus*, *Megasphaera*, and *Veillonella* in sputum samples [19].

A major conclusion from these studies is that Gram-negative microorganisms dominate in COPD patients’ lungs and there is a shift towards the increase of Gammaproteobacteria, which belong to the Proteobacteria phylum, and a decrease in Bacteroidetes phylum that dominates in healthy lungs. Moreover, it is evident that despite the existence of a core lung microbiome in COPD lungs, there are differences in the relative abundance of some genera in samples taken from the different compartments of lungs. BAL and sputum samples share a common pulmonary lung microbiome core in phyla; however, the Actinobacteria phylum is the most prevalent in BAL, while it is underrepresented in sputum samples as well as in samples that study normal lung microbiome. This difference was explained by Cabrera et al., who examined sputum and BAL samples as well as tissue samples, and concluded that the microbiome of the upper bronchial tree differs from that of the lower bronchial tree, as the latter has greater diversity [45].

## 9. Gut Microbiota in COPD Patients

As previously mentioned, the gut–lung theory suggests an interaction between the gut and lung microbiomes. In this direction, many patients with chronic lung disorders suffer from gastrointestinal symptoms [37]. At the same time, there is a higher probability (two to three times) of inflammatory bowel disease (IBD) diagnosis in patients with COPD. Furthermore, the lungs are affected in half of the patients with IBD and 33% with irritable bowel syndrome (IBS) [36]. These data support that the gut–lung axis is bidirectional, which was also confirmed by the studies of Dickson et al. and Sze et al. [51,52]. Sze et al. examined the effect of lipopolysaccharide-induced acute lung injury in murine lung, gut, and blood microbiomes. They found that the bacterial load in the cecum and blood was increased and proposed that the translocation of bacteria from the lungs to the bloodstream had taken place [51]. The transportation of microbes was also confirmed by Dickson et al., who examined mice with sepsis and humans with acute respiratory disease syndrome (ARDS). They found that the lung microbiome of both rats and humans was enriched with bacteria found in abundance in the gut that were transported via the portal circulation, systemic circulation, or mesenteric lymphatics [52]. According to studies that examined fecal samples in humans and murine models with COPD, the major phyla were Bacteroidetes, Firmicutes, Proteobacteria, Fusobacteria, Verrucomicrobia, and TM7 [5,8,53] and the major genera were Lactobacillus, Oscillospira, Clostridium, Ruminococcus, Blautia, Treponema, Allobaculum, and Turicibacter [5]. Bowerman et al. examined the feces of humans with COPD, and found that Bifidobacteriaceae, Eubacteriaceae, Lactobacillaceae, Micrococcaceae, Streptococcaceae, and Veillonellaceae were associated with COPD. At the same time, Desulfovibrionaceae, Bacteroidaceae Gastranaerophilaceae, and Selenomonadaceae were negatively associated with COPD [49]. Bacteroidetes, Proteobacteria, Lactobacillus, Oscillospira, Allobaculum, Treponema, and three species of *Streptococcus* were related to the development of COPD [5,49]. Chiu et al. found that the gut microbiome was not associated with the severity of COPD, but they pointed out that there were microbiota differences between the different stages of COPD. More specifically, it was found that the Ruminococcaceae NK4A214 group, Lachnoclostridium, and Bacteroidetes were more abundant in mild COPD (grade 1). At the same time, Tyzzerella and Dialister were less abundant in the stool samples of these patients. Furthermore, Fusobacterium and Aerococcus were more abundant in severe COPD (Stages 3 and 4) [48]. Another study by Lai et al. stated that Parabacteroides goldsteinii (Pg) and E.coli were inversely associated with COPD severity, while Lachnospiriaceae had a positive correlation with COPD severity [54]. *Bacteroides* spp. in stool samples has a negative correlation with the eosinophilic count, which is associated with the risk of COPD exacerbations, mortality, decreased FEV1, and the response to corticosteroids [48].

## 10. The Relation between Microbiomes and COPD Exacerbations

COPD exacerbations are acute events characterized by the worsening of respiratory symptoms [55]. A patient with COPD experiences 0.5 to 3.5 exacerbations yearly [3]. The primary cause of acute exacerbations of COPD (AECOPD) are respiratory infections (70–80%) caused predominantly by bacteria and viruses, and rarely by fungi. However, environmental factors can initiate an acute exacerbation [4,56]. The causal factor classifies exacerbations as either bacterial or viral and according to the level of eosinophil count in blood samples [50]. This classification is very important as it can help administer the proper treatment, given that eosinophilic exacerbations are responsive to inhaled corticosteroids [53]. 

The five most prevalent phyla in sputum samples from patients with COPD during exacerbations are Firmicutes, Actinobacteria, Bacteroidetes, Proteobacteria, and Fusobacteria [12,33,57], while in the fecal samples of mice, the most prevalent phyla are Firmicutes, Bacteroidetes, Proteobacteria, and TM7 [5]. The most dominant genera in sputum samples during acute exacerbations of COPD are *Streptococcus*, which is the most prevalent, *Neisseria*, *Porphyromonas*, *Haemophilus*, *Veillonella*, *Prevotella*, *Rothia*, *Pseudomonas*, *Staphylococcus*, *Proteus*, and *Moraxella* [12,19,33,57,58,59]. Studies have shown that reduced microbial diversity is associated with COPD exacerbation events [19,31,33,57,59]. 

There is evidence that the increase of a taxa species can induce the rise of organisms of the same taxa that altogether contribute to exacerbations [36,60]. In the lung microbiome, the increase in Proteobacteria and Actinobacteria and decrease in Firmicutes and Bacteroidetes are related to the onset of exacerbations [31,33,57,59,60,61]. In sputum samples of AECOPD patients, the proportions of Staphylococcus, Moraxella, *Haemophilus*, *Rothia*, and *Stenotrophomonas* are increased, and those of *Lactobacillus*, *Streptococcus*, *Veillonella*, *Prevotella*, *Alloprevotella*, and *Porphyromonas* are decreased compared to stable COPD [33,50,57,61] In fecal samples, AECOPD is positively associated with an increase in the abundance of Firmicutes, Ruminococcus, Blautia, Clostridium, and Turicibacter [5]. Decreased microbiome diversity, decrease in Streptococcus, and dominance of Proteobacteria (mainly *Haemophilus* spp.) can lead to bacterial exacerbations, while the decreased Proteobacteria/Firmicutes ratio and increased Firmicutes are found in eosinophilic exacerbations [7,50,57]. The length of stay (LOS) in the hospital during an exacerbation is positively related with the presence of *Staphylococcus*, *Achromobacter*, *Pseudomonas*, *Stenotrophomonas*, and *Ralstonia* in sputum samples, whereas the presence of *Veillonella*, *Peptostreptococcus*, *Fusobacterium*, and *Porphyromonas* are inversely correlated with the LOS [12,15]. Another critical element is the microbiome of sputum samples on the first day of hospitalization due to AECOPD, which can predict the mortality risk. Higher microbial diversity in sputum samples is correlated with a positive outcome after admission due to AECOPD, and the presence of Veillonella in sputum samples is positively associated with a decreased mortality, while the presence of Staphylococcus is correlated with an increased risk of mortality [12]. These results may be explained by Su et al., who found a positive correlation between Veillonella and lung function (FEV1% and FEV1/FVC) and between Staphylococcus and CRP [33]. In this same study, Alloprevotella was negatively correlated with CRP [33]

Another important element is to predict the possibility of frequent exacerbations. According to studies, it was found that *Haemophilus*, *Moraxella*, *Proteobacteria*, and decreased microbiome diversity are related to the high frequency of exacerbations, while *Lactobacillus* and *Streptococcus* have a negative correlation with frequent exacerbations [42,50,58,61,62]. Moreover, Wang et al. suggested that the microbiome in a stable state does not predict exacerbation frequency [19].

The studies that examined the association between lung and gut microbiomes and the acute exacerbation of COPD are summarized in Table 2.

## 11. Lung Function and Microbiome

There is an association not only between the human microbiome and the pathogenesis of COPD, but also with lung function. Erb-Downward et al. examined the lung tissue obtained from six patients who underwent lung transplants due to advanced COPD, and found that the presence of *Pseudomonas* spp. was associated with the impairment of lung function [35]. Other studies correlated Proteobacteria, reduced microbiome diversity, *Neisseria*, and *Haemophilus* with deterioration in lung function, COPD severity, and mortality. In contrast, Firmicutes, except for *Streptococcus* isolated in sputum, were related to increased disease severity, whereas Prevotella, Veillonella, and Actinomyces were negatively associated with COPD severity [29,30,31,44,58]. The presence of *Bacteroides* spp., Desulfovibrio piger_A, and CAG-302 sp001916775 was positively correlated with lung function [47,48]. In stool samples, Bacteroidetes and Alloprevorella isolation during a year contributed to little or no pulmonary function reduction during the same period [48]. Streptococcus sp000187445 and *S. Vestibularis* were negatively correlated with lung function (FEV1) [49]. In stool samples, Firmicutes were related to a decline in lung function for the one year period. Whether the presence of Prevotella was associated with better lung function is a matter of debate. Some studies suggest a possible pathogenic role [47]. On the contrary, other studies showed no or only partial benefit when Prevotella species were abundant [56].

## 12. The Potential Impact of Current Knowledge on Microbiomes in the Treatment of COPD

Although our knowledge regarding the role of gut and lung dysbiosis in the pathogenesis of COPD and COPD exacerbations is limited, studies have shed some light on the potential role that several treatment options might have in preventing and treating COPD. These treatment options, such as prebiotics or postbiotics, diet modification, and administration of drugs, such as steroids and antibiotics could affect the lung or gut microbiomes.

Lai et al. examined the effects of the administration of antibiotics (vancomycin, neomycin metronidazole, and ampicillin, and a combination of these) and fecal microbiota transplantation (FMT) in mice with COPD. They found that treatment with ampicillin and vancomycin, as well as combinational antibiotic treatment improved the clinical status of mice. Then, fecal microbiota from these mice or healthy donors was administered in COPD mice, and this resulted in an improvement of COPD symptoms. It was understood that the beneficial effects of antibiotics and FMT were due to a reduction in intestinal inflammation, and promotion of intestinal integrity, as both contributed to the amelioration of COPD pathogenesis [54].

A diet poor in fiber can cause gut dysbiosis and induce local and systemic chronic inflammation [8,13], while a Western diet has been negatively associated with survival [21]. Moreover, fiber-rich diets, and diets rich in antioxidants have been associated with better lung function, lowering the risk of COPD, and decreasing mortality [8,21]. This indicates that high fiber intake might be essential to preventing and managing COPD [8,13,49].The positive impact of fiber-rich diets is attributed to the ability of dietary fibers to change the gut microbiome by altering the Firmicutes to Bacteroidetes ratio, allowing for the outgrowth of Bacteroidetes that are capable of fermenting high amounts of SCFAs. The role of SCFAs has been previously described [37]. Jang et al. found that fecal microbiota therapy (FMT), a high fiber diet (containing 20% fibers), and the combination of the two attenuated emphysema development in mice by inhibiting local and systemic inflammation, and changing the composition of the gut and lung microbiomes. More specifically, in the fecal samples, the abundance of the Bacteroidetes phylum was increased in addition to the increase of the Lachnospiraceae and Bacteroidaceae families, while the Firmicutes/Bacteroidetes (F/B) ratio was reduced in both lung and fecal samples and the abundance of the Lactobacillaceae family was decreased in fecal samples [63]. The positive effect of the increase of abundance of Lachnospiriaceae is confirmed by Lai et al. who found that Lachnospiriaceae contributed to the improvement of COPD symptoms [54]. This could be explained by the fact that the increase in the abundance of the Lachnospiraceae and Bacteroidaceae family resulted in an increase in the concentration of SCFAs as bacteria of these two families metabolize dietary fibers into SCFAs [63]. Furthermore, Jang et al. found that a high-fiber diet was associated with better lung function and prevented emphysema compared with postbiotic administration of SCFAs. This change can be attributed to the fact that a fiber-rich diet results in the production of many metabolites and not only SCFAs [63].

Li et al. examined the impact of Western medicine (moxifloxacin hydrochloride tablets + salbutamol sulfate tablets), and integrated Chinese and Western medicine (Tong Sai granules + moxifloxacin hydrochloride tablets + salbutamol sulfate tablets + Bu Fei Yi Shen granules + salbutamol sulfate tablets), on the intestinal microbiota of COPD rats with acute exacerbation. They found that Western medicine increased the Firmicutes/Bacteroidetes ratio, suggesting that it regulates the intestinal microbiota, as this ratio has been associated with maximum oxygen consumption, improving the hypoxic state of exacerbation in COPD. The combination of Chinese and Western medicine improved lung function by reducing the pathological damage in pulmonary alveoli and decreasing systemic and intestinal inflammation. In fecal samples, administration of Western medicine increased Firmicutes, Lactobacillus, Allobaculum, Ruminococcus, Blautia, and Treponema and decreased Bacteroidetes, Proteobacteria, TM7, Oscillospira, Clostridium, and Turicibacter. Conversely, combining Chinese and Western medicine has resulted in increasing Bacteroidetes, Proteobacteria, Lactobacillus, and Allobaculum and decreased Firmicutes, TM7, Blautia, Treponema, Oscillospira, Clostridium, Ruminococcus, and Turicibacter [5].

Probiotics benefit humans due to their antimicrobial activity and ability to enhance the gut epithelium barrier and immunodulation [3,8]. Administration of probiotic bacteria such as Lactobacillus rhamnosus, and Bifidobacterium breve can suppress immune response caused due to exposure to cigarette smoke and consequently COPD [64]. As previously mentioned Lai et al. found out that *Parabacteroides goldsteinii* (Pg) was inversely associated with disease severity in COPD mice. They also examined the effect of oral administration of Pg MTS01 and Pg-LPS in these mice. They found that both contribute to the improvement of Cigarette smoke-induced COPD syndromes, and therefore they can be used as potential therapeutic agents in the treatment of COPD [54]. There are contrasting results on the effect of steroids and antibiotics frequently used in treating COPD and COPD exacerbations in microbiome diversity. Some studies showed that steroid use increases microbiome diversity, and antibiotic usage reduces bacterial diversity [38,60]. At the same time, other studies showed that steroids are associated with reduced microbiome diversity, and antibiotics with increased diversity [57]. Treatment with only oral systemic steroids increases Proteobacteria, Bacteroidetes, and Firmicutes phyla, increases the Proteobacteria/Firmicutes ratio, *Haemophilus* and *Moraxella* abundance [57,60], while treatment with only antibiotics reduces Proteobacteria members. It should be noted that the changes caused by antibiotic administration can be reversed more quickly by the co-administration of corticosteroids [60]. It must be pointed out that these treatments have a prolonged effect on the lung microbiome [57,60]. An antibiotic that has been widely studied is azithromycin, which influences the systemic inflammation directly via its impact on host immune cells and indirectly via alteration of the gut [65] and lung microbiomes [66,67]. It reduces microbiome diversity [67] and increases the abundance of Anaerococcus, lowering that of *Haemophilus*, *Pseudomonas*, and *Staphylococcus* [66]. Azithromycin also reduces the frequency of exacerbations by decreasing mucus secretion and by stimulating the production of SCFAs that have immunomodulatory effects [17,66]. Moreover, it can improve gastric emptying and reduce gastroesophageal reflux, a risk factor for increased COPD exacerbation frequency [68]. It also attenuates emphysema-like changes and lung function loss [69]. However, long-term use of azithromycin can result in strains of bacteria resistant to azithromycin, thereby making the future treatment more challenging [68].

The right choice of antibiotics is of great importance in the exacerbations of COPD. The wrong choice can induce antibiotic resistance, resulting in high rates of clinical failure and increasing the cost of treatment. Therefore, it is necessary to find indicators for treatment success or failure [15]. Liu et al. assessed the relationship between sputum microbiome during exacerbation of COPD and its response to empiric antibiotic treatment. They found that an increase in Proteobacteria taxa and a decrease in non-Proteobacteria taxa, as well as the microbiome diversity were associated with the failure of antibiotic therapy. Furthermore, antibiotic treatment failure has been associated with Pseudomonas, Achromobacter, Stenotrophomonas, and Ralstonia in sputum samples during the exacerbation state. Moreover, Prevotella, Peptostreptococcus, Leptotrichia, and Selenomonas have been associated with the success of antibiotic treatment. The fact that those with similar microbiome profiles responded to the same adjusted therapy revealed that the sputum microbiome at the time of hospitalization might indicate which patients will respond to the adjusted therapy [15].

The studies that examined the treatment of COPD that target lung and gut microbiomes are summarized in Table 3.

## 13. Conclusions

The association between lung and gut microbiomes in the pathogenesis of COPD has been recently uncovered, and this knowledge has shed light on the pathophysiology of COPD. It is evident that the lung and gut microbiomes affect each other and both play a vital role in the pathogenesis of COPD. However, more research needs to be carried out to find the exact associations between the microorganisms and COPD pathophysiology and exacerbation genesis. Another field that research should focus on is the impact of treatment interventions targeting the human microbiome in preventing COPD and COPD progress. Further research is needed to examine a larger sample of COPD patients with a different COPD severity status.

## Figures and Tables

**Table 1 jpm-13-00804-t001:** Studies examining the relationship between the lung and gut microbiota with COPD pathophysiology.

Study	Sample	Human/Rats	Study Sample Size	Conclusions
(1) Sze et al., 2012 [39]	Lung tissue	Human	Thirty-two (eight smokers, eight non-smokers without COPD, eight with COPD GOLD 4, and four with CF).	Increase of Firmicutes, *Lactobacillus*, and *Burkholderia* species in COPD compared to other groups.
(2) Pragman et al., 2018 [40]	Lung tissue, nasal, brochial, oral	Human	Nine patients( three with mild and six with moderate COPD).	*Streptococcus* was the most common genus in COPD. Upper resperatory brochial tree is richer but less diverse in microorganisms than the lower brochial tree.
(3) John R. Erb-Downward et al., 2011 [35]	BAL	Human	Fourteen (seven healthy smokers, four smokers with COPD, thee non-smokers).	Low microbial diversity and presence of *Pseudomonas* spp. were associated with a decrease in lung function and progression of COPD.
	Lung tissue	Human	Eight specimens from six COPD patients.	(1) Differences in the microbiome in different regions of the same lung; (2) Low microbial diversity and presence of *Pseudomonas* spp. associated with a decrease in lung function.
(4) Sze et al., 2015 [41]	Lung tissue	Human	Forty samples from five COPD patients with GOLD 4 and 28 samples from four healthy donors.	(1) Low microbiome diversity was associated with the pathogenesis of COPD; (2) Increase of Proteobacteria and Actinobacteria and a decrease in Firmicutes and Bacteroidetes phyla was observed in COPD patients.
(5) Dicker et al., 2021 [42]	Sputum	Human	Two hundred and fifty-two cases of stable COPD.	(1) Proteobacteria dominance was associated with the impairment of lung function and frequency of exacerbations; (2) Low microbiome diversity and high abundance of Proteobacteria and *Haemophilus* were associated with eosinopenia, while high abundance of Firmicutes and *Streptococcus* was associated with eosinophilia.
(6) Galiana et al., 2014 [43]	Sputum	Human	Nineteen (nine moderate or mild COPD patients, 10 severe or very severe COPD patients).	*Actinomyces* was associated with COPD severity.
(7) Marian Garcia-Nuñez et al., 2014 [44]	Sputum	Human	Seventeen COPD patients.	(1) Low microbial diversity was associated with progression of COPD; (2) Advanced disease was associated with the increase of Proteobacteria phylum and decrease of Firmicutes phylum.
(8) Wang et al., 2019 [19]	Sputum	Human	One hundred and one samples from 16 healthy subjects and 43 COPD patients.	(1) Increased relative abundance of *Moraxella*, *Streptococcus*, and Actinobacteria, and a low microbial diversity in COPD patients; (2) *Haemophilus* and *Neisseria* were positively associated with sputum neutrophil counts, and *Streptococcus*, *Megasphaera*, and *Veillonella* was negatively associated with sputum neutrophil counts.
(9) Wu et al., 2014 [6]	Sputum	Human	Twenty (10 healthy controls and 10 COPD patients).	(1) Increased diversity in COPD patients; (2) Increase of *Streptococcus pneumoniae*, *Klebsiella pneumoniae*, and *Pseudomonas aeruginosa* in COPD patients.
(10) Zakharkina et al., 2013 [30]	BAL	Human	Eighteen (nine heathy controls, nine COPD patients).	(1) The phylum Cyanobacteria and genera *Afipia*, *Brevundimonas*, *Curvibacter*, *Moraxella*, *Neisseria*, *Undibacterium*, *Corynebacterium*, *Capnocytophaga*, and *Leptolyngbia* were characteristic for COPD.
(11) Pragman et al., 2012 [25]	BAL	Human	Thirty-two (22 patients with moderate/severe COPD, 10 healthy controls).	Increase in microbial diversity was associated with the development of COPD.
(12) Seixas et al., 2021 [32]	BAL	Human	One hundred and six patients with lung disease, seven of whom suffer from COPD GOLD 2.	Association of *Haemophilus* with COPD.
(13) Cabrera et al., 2012 [45]	BAL, sputum	Human	Six moderate COPD patients.	Sputum has lower microbial diversity compared with that of BAL and lung tissue.
(14) Molyneaux et al., 2013 [46]	Sputum	Human	31 (14 COPD and 17 healthy)	Increase in Proteobacteria and Veillonellaceae and decrease in Firmicute in COPD.
(15) Chiu et al., 2022 [47]	Feces	Human	Fifty-five COPD patients.	Bacteroidetes and *Alloprevorella* during a one-year period contributed to little or no reduction in lung function. During the same period, Firmicutes were related to a decline in lung function.
(16) Chiu et al., 2021 [48]	Feces	Human	Sixty COPD patients.	(1) Gut microbiome was not associated with severity of COPD; (2) *Ruminococcaceae NK4A214 group*, *Lachnoclostridium*, and Bacteroidetes were more abundant in mild COPD, while *Tyzzerella 4* and *Dialister* were less abundant; (3) *Fusobacterium* and *Aerococcus* were more abundant in severe COPD; (4) *Bacteroides* sp. had a positive correlation with lung function and a negative correlation with eosinophil count.
(17) Bwerman et al., 2020 [49]	Feces	Human	Fifty-seven (28 COPD patients and 29 healthy controls).	(1) *Bifidobacteriaceae*, *Eubacteriaceae*, *Lactobacillaceae*, *Micrococcaceae*, *Streptococcaceae*. and *Veillonellaceae* increased in COPD while *Desulfovibrionaceae*, *Gastranaerophilaceae*. and *Selenomonadaceae decreased*; (2) *S. vestibularis* and *two unnamed Streptococcus species (sp001556435*, *sp000187445)* were enriched in COPD; (3) Lung function had a positive correlation with *Desulfovibrio piger_A* and *CAG-302 sp001916775* and a negative correlation with *Streptococcus sp000187445* and *S. vestibularis*.

Abbreviations: BAL, bronchoalveolar lavage; CF, cystic fibrosis; COPD, chronic obstructive pulmonary disease.

**Table 2 jpm-13-00804-t002:** Studies investigating the relation between lung and gut microbiomes and acute COPD exacerbations.

Study	Sample	Human/Rats	Study Sample Size	Conclusions
(1) Leitao et al., 2019 [12]	Sputum	Human	One hundred and two patients.	High microbial diversity and the presence of Veillonella was associated with a positive outcome and a shorter hospital LOS, while the presence of Staphylococcus was associated with a longer LOS and increased mortality risk.
(2) Wang et al., 2019 [19]	Sputum	Human	One hundred and one sputum samples from 16 healthy subjects and 43 COPD patients.	(1) During COPD exacerbations, increased Moraxella and decreased microbial diversity was observed compared to a stable state; (2) The microbiome at a stable state did not predict exacerbation frequency.
(3) Su et al., 2022 [33]	Sputum	Human	Seventy-six samples from 28 patients with AECOPD, 23 stable COPD, 15 in recovery, and 10 healthy controls.	(1) Low microbial diversity was associated with AECOPD; (2) Decrease in Firmicutes and Bacteroidetes and increase in Proteobacteria and Actinobacteria was found in AECOPD patients; (3) Increased proportions of Rothia, unidentified Corynebacteriaceae, and Stenotrophomonas, and decreased levels of Prevotella, Alloprevotella, *Porphyromonas*, and unidentified Prevotellaceae were found in AECOPD patients; (4) Positive correlation between Veillonella and lung function and negative association between *Haemophilus* and *Prevotella* with a severity of dyspnea; (5) CRP levels were positively associated with Staphylococcus and negatively correlated with Alloprevotella.
(4) Mayhew et al., 2018 [50]	Sputum	Human	Five hundred and eighty-four samples from 101 COPD patients.	Moraxella was associated with increased risk for exacerbations, and Lactobacillus was negatively correlated with exacerbation frequency.
(5) Wang et al., 2016 [57]	Sputum	Human	Four hundred and seventy-six samples from 87 COPD patients (stable state, exacerbation, 2 weeks post-therapy, and 6 weeks recovery).	Low microbial diversity, increase in Proteobacteria and Moraxella, and a decrease in Firmicutes were observed during exacerbations.
(6) Huang et al., 2014 [60]	Sputum	Human	Sixty samples from 12 patients (before, at onset, after an exacerbation).	(1) Members of the Proteobacteria phylum were increased in exacerbations; (2) An increase in taxa species can induce the rise of organisms of the same taxa that altogether contribute to exacerbation pathogenesis.
(7) Liu et al., 2020 [15]	sputum	Human	Forty-one AECOPD patients and 26 healthy controls.	LOS in hospital during an exacerbation was positively correlated with the presence of Achromobacter, Pseudomonas, Stenotrophomonas, and Ralstonia in sputum samples, whereas the presence of Peptostreptococcus, Fusobacterium, and *Porphyromonas* was inversely correlated with LOS.

Abbreviations: AECOPD, acute exacerbation of COPD; CRP, C-reactive protein; LOS, length of stay.

**Table 3 jpm-13-00804-t003:** Studies regarding the impact of treatment targeting microbiomes in COPD pathophysiology.

Study	Sample	Human/Rats	Study Sample Size	Conclusions
(1) Li et al., 2021 [5]	Fecal samples, lung, and intestinal tissues	Rats	25	Intervention of integrated Chinese and Western medicine improved lung function, reduced the pathological injury of alveoli and intestines, and alleviated the systematic inflammatory response.
(2) Jang et al., 2020 [63]	Feces	Mouse	88	(1) FMT and HFD and their combination attenuated emphysema development; (2) FMT and HFD altered the gut microbiome composition.
(3) Lai et al. 2020, [54]	Feces	Rats	Not mentioned	FMT and administration of *Parabacteroides goldesteinii* (Pg) and Pg-Lps relieved COPD symptoms.
(4) Wang et al., 2016 [57]	Sputum	Human	Four hundred and seventy-six samples from 87 COPD patients (stable state, exacerbations, 2 weeks post-therapy, and 6 weeks recovery).	(1) Treatment with corticosteroids resulted in decreased microbial diversity, an increase of Proteobacteria over Firmicutes, a decrease of Streptococcus, and an increase of *Haemophilus* and *Moraxella*; (2) An opposite trend in both the diversity and microbial composition changes was observed for subjects treated with antibiotics.
(5) Huang et al., 2014 [60]	Sputum	Human	60	Treatment with antibiotics decreased the abundance of Proteobacteria. Treatment with corticosteroids led to an enrichment of Proteobacteria phyla.
(6) Segal et al., 2017 [67]	BAL	Human	Twenty smokers (current or ex-smokers) with emphysema.	Azithromycin had an anti-inflammatory effect via the induction of bacterial metabolites and reduced microbial diversity.
(7) Slater et al., 2014 [66]	Saline washings	Human	Five patients with moderate/severe asthma.	Azithromycin increased the abundance of *Anaerococcus* while lowering the quantity of *Haemophilus*, *Pseudomonas*, and *Staphylococcus*.
(8) Macowan et al., 2020 [69]	Lung tissue	Rats	fifteen (10 in cigarette smoke and 5 in fresh air).	Azithromycin attenuated emphysematous changes due to smoke exposure.
(9) Liu et al., 2020 [15]	Sputum	Human	Forty-one AECOPD patients and 26 healthy controls.	(1) Increase in Proteobacteria, the presence of *Pseudomonas*, *Achromobacter*, *Stenotrophomonas*, and *Ralstonia*, and the poor microbial diversity was associated with the failure of antibiotic therapy; (2) The presence of *Prevotella*, *Peptostreptococcus*, *Leptotrichia*, and *Selenomonas* was associated with the success of antibiotic treatment; (3) Patients with similar microbiome profiles responded to the same adjusted therapy after treatment failure.

Abbreviations: AECOPD, acute exacerbations of COPD; BAL, bronchoalveolar lavage; FMT, fecal microbiota transplantation; HFD, high-fiber diet (20% fiber); integrated Chinese and Western medicine (Tong Sai granules + moxifloxacin hydrochloride tablets + salbutamol sulfate tablets + Bu Fei Yi Shen granules + salbutamol sulfate tablets).

## Data Availability

Not applicable.

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
