# Peer review of "Lung and Gut Microbiome in COPD"

_jpm, 2023, doi:10.3390/jpm13050804_

Round 1

Reviewer 1 Report

This manuscript reviews the relationship between COPD and the microbiome in lungs and gut.

The language in the manuscript is very clear and well written and only very small typos could be seen. The article is overall very pleasing to the reader and the subject is current and relevant to the research community.

One major review that could improve the quality of this manuscript is:

*The tables 1 and 2 summarize well the findings in different tissues, contexts and associations. Because of that, the text that accompanies them feels redundant as it directly enunciates the same findings. It would be great if instead, different comparisons were drawn (for example, between different sampling: what is different and similar between BAL x tissue x sputum and what could be an explanation; or how contradictory findings could be possibly explained) and more discussion was included instead of listing. Another possibility would be to discuss the most important phyla that came out in these findings, so to make it informative for non-microbiologists that are reading the review can understand the possible meaning of those findings. Pointing out in the tables or text about the usage of antibiotics and corticosteroids among the studies' subjects could also add interesting insight.

*Another important comment is that the citation of the animal studies in the tables may not be advisable, as it adds very little value to the manuscript, and have very disputable meaning as most COPD murine models only emulate partially the components of human COPD and can hardly be called COPD, as well as it is hard to compare the different species microbiome in very different dietary conditions.

*Furthermore, please refrain from citing reviews instead of the original publications for impacting affirmations such as in the paragraph starting at line 381, or in line 133, when data for proportion of smokers that develop COPD is only cited in the  reference 38 and their citations are over 50 years old.  

Regarding small typos and very minor corrections, a few that I could detect are listed below:

Line 82: scientific notation

Line 143: citation 35

Table 1, par 15: inappropriate italic

Line 162: transplant to

Line 228: severity

Line 284: not clear about eosinophil count (is it blood?), phrasing

Line 288: period after Firmicutes

Author Response

REVIEWER 1

This manuscript reviews the relationship between COPD and the microbiome in lungs and gut. The language in the manuscript is very clear and well written and only very small typos could be seen. The article is overall very pleasing to the reader and the subject is current and relevant to the research community.

Response: We sincerely thank you for your kind words about our paper. We are delighted to receive positive feedback from you.

One major review that could improve the quality of this manuscript is:

  1. *The tables 1 and 2 summarize well the findings in different tissues, contexts and associations. Because of that, the text that accompanies them feels redundant as it directly enunciates the same findings. It would be great if instead, different comparisons were drawn (for example, between different sampling: what is different and similar between BAL x tissue x sputum and what could be an explanation; or how contradictory findings could be possibly explained) and more discussion was included instead of listing.

RESPONSE: We really thank you for this point. We have revised the manuscript according to your suggestions.

Another possibility would be to discuss the most important phyla that came out in these findings, so to make it informative for non-microbiologists that are reading the review can understand the possible meaning of those findings. Pointing out in the tables or text about the usage of antibiotics and corticosteroids among the studies' subjects could also add interesting insight.

RESPONSE: We really thank you for this point. We have revised the manuscript according to your suggestions. Moreover neither antibiotics nor corticosteroids have been used amongst studies subjects.

  1. *Another important comment is that the citation of the animal studies in the tables may not be advisable, as it adds very little value to the manuscript, and have very disputable meaning as most COPD murine models only emulate partially the components of human COPD and can hardly be called COPD, as well as it is hard to compare the different species microbiome in very different dietary conditions

Response: We thank you for the remark. We have removed animal studies from the tables.

  1. *Furthermore, please refrain from citing reviews instead of the original publications for impacting affirmations such as in the paragraph starting at line 381, or in line 133, when data for proportion of smokers that develop COPD is only cited in the reference 38 and their citations are over 50 years old.  

Response: Thank you for your comment. We have cited the original articles and have removed the data for proportion of smokers that develop COPD.

  1. Regarding small typos and very minor corrections, a few that I could detect are listed below: Line 82: scientific notation Line 143: citation 35 Table 1, par 15: inappropriate italic Line 162: transplant to Line 228: severity Line 284: not clear about eosinophil count (is it blood?), phrasing Line 288: period after Firmicutes

Response: Thank you for your comments. The minor changes have been made,as suggested.

We appreciate you taking the time to offer us your insights related to the paper. We found your feedback very constructive. We tried to be responsive to your concerns.

Reviewer 2 Report

The authors have outlined a review highlighting the importance of the microbiome on the disease COPD. The review includes tables summarizing what is known about the bacteria isolated from lung tissue, BALF, sputum and feces. However, the review article fails to deliver on its title which eludes to a discussion about the coexistence or codestruction(sp?) of the microbiome in COPD. Another major concern about this review is it fails to expand on what is already published in existing review articles about COPD and the microbiome.

Concerns listed below highlight the need to provide more in depth evaluation of the current literature.  

1 - References do not match. Example Table 1 lists reference 5 as Li et al 2021 investigating rats, but reference 5 is listed multiple times in the text (lines 252 and 290) stating human studies. This should be reconciled.

2 - Table 1 is missing many studies of the microbiome in COPD patients. Other review articles have provided a more comprehensive list and description of this subject matter.

3 - It would be valuable to expand on the vicious circle hypothesis and discuss how the microbiome is associated with influencing lung function. Instead of a table with the known microbiome of patient samples, correlate these findings with lung function.

4 - Do the studies observing the microbiome in feces of COPD patients discuss the status of these patients and whether they have GI-disease? Do they have IBS or IBD? If they do does this alter their lung microbiome or gut microbiome and subsequent lung function?

5 - The back and forth between human and rodent should be made more clear. Recommend removing rodent information from the same table as humans. Furthermore, do the rat and human studies align or is their conflicting data between the two? If so explain why. Are rodents a good model for COPD and microbiome studies?

6 - Lines 187-190 states several bacteria that are part of the phylum Cyanobacteria and associated with COPD patients. This seems wrong, please correct.

7 - Lines 194-197 identify studies with contradictory results. This warrants more discussion.

8 - Would be interesting to expand on the potential use of FMT as a treatment for COPD. Interesting article by Lai et al. PMID 33687943

9 - Are there any conclusions about effective antibiotics that can be made? Best or worse to use?

Author Response

REVIEWER 2

The authors have outlined a review highlighting the importance of the microbiome on the disease COPD. The review includes tables summarizing what is known about the bacteria isolated from lung tissue, BALF, sputum and feces.

  1. However, the review article fails to deliver on its title which eludes to a discussion about the coexistence or codestruction (sp?) of the microbiome in COPD. Another major concern about this review is it fails to expand on what is already published in existing review articles about COPD and the microbiome.

Concerns listed below highlight the need to provide more in depth evaluation of the current literature.  

1 - References do not match. Example Table 1 lists reference 5 as Li et al 2021 investigating rats, but reference 5 is listed multiple times in the text (lines 252 and 290) stating human studies. This should be reconciled

RESPONSE: Thank you for the comments. We have stated in the text that reference 5 (Li et al) is a rat study and not a human study.

2 - Table 1 is missing many studies of the microbiome in COPD patients. Other review articles have provided a more comprehensive list and description of this subject matter.

RESPONSE: Thank you for this remark. We have revised the manuscript according to your suggestions and introduced more studies regarding the role of microbiome in COPD patients.

3 - It would be valuable to expand on the vicious circle hypothesis and discuss how the microbiome is associated with influencing lung function. Instead of a table with the known microbiome of patient samples, correlate these findings with lung function.

RESPONSE: Thank you for the comments. We have revised the manuscript according to your suggestions.

4 - Do the studies observing the microbiome in feces of COPD patients discuss the status of these patients and whether they have GI-disease? Do they have IBS or IBD? If they do does this alter their lung microbiome or gut microbiome and subsequent lung function?

RESPONSE: Thank you for the comment. In these studies, the investigators did not mention if the patients have comorbidities from the GI tract apart from study 46 that it mentioned that participants were excluded if they have co-morbidities (such as IBD, IBS) with an altered microbiome.

5 - The back and forth between human and rodent should be made more clear. Recommend removing rodent information from the same table as humans. Furthermore, do the rat and human studies align or is their conflicting data between the two? If so explain why. Are rodents a good model for COPD and microbiome studies?

RESPONSE: Thank you for these points. We have removed rodent data from the tables.

6 - Lines 187-190 states several bacteria that are part of the phylum Cyanobacteria and associated with COPD patients. This seems wrong, please correct.

RESPONSE: Thank you for this remark. We have changed it appropriately

7 - Lines 194-197 identify studies with contradictory results. This warrants more discussion.

RESPONSE: Thank you for your comment. We have discussed more these contradictory results.

8 - Would be interesting to expand on the potential use of FMT as a treatment for COPD. Interesting article by Lai et al. PMID 33687943

RESPONSE: Thank you for the comments. We have revised the manuscript according to your suggestions.

9 - Are there any conclusions about effective antibiotics that can be made? Best or worse to use?

RESPONSE: Thank you for your comments. Azithromycin is a very effective antibiotic that should be used in the treatment of COPD patients.

We appreciate you taking the time to offer us your insights related to the paper. We found your feedback very constructive. We tried to be responsive to your concerns.

Round 2

Reviewer 2 Report

The authors failed to address my concern about the title not representing the material provided in the review. The title also still has a misspelled word, codestrunction. There is no discussion about coexistence or codestruction and should not be part of the title.

Line 31: replace "accuse" with "be the cause of"

Line 52: reword "moreover, cigarette..."

Line 60: replace "by" with "with"

Line 81: what is mean by the gut microbiome is the richest in the microorganism population?

Line 90: replace "takes place" with "is involved"

Line 102: add "of the body" after total microbiome

Line 125: reword

Line 152: Table 1 and associated text. Be consistent with using either Gold IV or Gold 4, but not both.

Line 174: states 10 species, but on lines 176-178 only lists 9 species.

Line 196: change BAL to BALF.

Line 207: replace "reduce" with "reduction"

Line 249: replace "consumption" with "conclusion"

Line 315: correct 0,5 and 3,5 to correct numbers.

Line 316: remove 3 

Author Response

The authors failed to address my concern about the title not representing the material provided in the review. The title also still has a misspelled word, codestrunction. There is no discussion about coexistence or codestruction and should not be part of the title.

RESPONSE: We have removed the word of coexistence and codestrunction from the title

Line 31: replace "accuse" with "be the cause of"

Line 52: reword "moreover, cigarette..."

Line 60: replace "by" with "with"

Line 81: what is mean by the gut microbiome is the richest in the microorganism population?

Line 90: replace "takes place" with "is involved"

Line 102: add "of the body" after total microbiome

Line 125: reword

Line 152: Table 1 and associated text. Be consistent with using either Gold IV or Gold 4, but not both.

Line 174: states 10 species, but on lines 176-178 only lists 9 species.

Line 196: change BAL to BALF.

Line 207: replace "reduce" with "reduction"

Line 249: replace "consumption" with "conclusion"

Line 315: correct 0,5 and 3,5 to correct numbers.

Line 316: remove 3 

Response: Thank you for your comments. The changes have been made, as suggested.

We appreciate you taking the time to offer us your insights related to the paper. We found your feedback very constructive. We tried to be responsive to your concerns.